nanotechnology

magnetic wire, nanoparticle assembly, parallel arrays

**Author for correspondence:**
Takashi Ogi
e-mail: ogit@hiroshima-u.ac.jp

This article has been edited by the Royal Society of Chemistry, including the commissioning, peer review process and editorial aspects up to the point of acceptance.

# Enhanced magnetic performance of aligned wires assembled from nanoparticles: from nanoscale to macroscale

Qing Li[1], Christina W. Kartikowati[2], Toru Iwaki[3], Kikuo Okuyama[3] and Takashi Ogi[3]

[1]Department of Environmental Science and Engineering, Fudan University, Shanghai 200433, People's Republic of China
[2]JurusanTeknik Kimia, FakultasTeknik, Universitas Brawijaya, Jl. MT. Haryono 167, Malang 65145, Indonesia
[3]Department of Chemical Engineering, Graduate School of Engineering, Hiroshima University, 1-4-1 Kagamiyama, Higashi, Hiroshima 739-8527, Japan

QL, 0000-0003-0587-1748; CWK, 0000-0001-5288-0378; KO, 0000-0002-1477-1442; TO, 0000-0003-3982-857X

Magnetic wires in highly dense arrays, possessing unique magnetic properties, are eagerly anticipated for inexpensive and scalable fabrication technologies. This study reports a facile method to fabricate arrays of magnetic wires directly assembled from well-dispersed $\alpha''$-Fe$_{16}$N$_2$/Al$_2$O$_3$ and Fe$_3$O$_4$ nanoparticles with average diameters of 45 nm and 65 nm, respectively. The magnetic arrays with a height scale of the order of 10 mm were formed on substrate surfaces, which were perpendicular to an applied magnetic field of 15 T. The applied magnetic field aligned the easy axis of the magnetic nanoparticles (MNPs) and resulted in a significant enhancement of the magnetic performance. Hysteresis curves reveal that values of magnetic coercivity and remanent magnetization in the preferred magnetization direction are both higher than that of the nanoparticles, while these values in the perpendicular direction are both lower. Enhancement in the magnetic property for arrays made from spindle-shape $\alpha''$-Fe$_{16}$N$_2$/Al$_2$O$_3$ nanoparticles is higher than that made from cube-like $\alpha''$-Fe$_{16}$N$_2$/Al$_2$O$_3$ ones, owing to the shape anisotropy of MNPs. Furthermore, the assembled highly magnetic $\alpha''$-Fe$_{16}$N$_2$/Al$_2$O$_3$ arrays produced a detectable magnetic field with an intensity of approximately 0.2 T. Although high-intensity external field benefits for the fabrication of magnetic arrays, the newly developed technique provides an environmentally friendly and feasible approach to fabricate magnetic wires in highly dense arrays in open environment condition.

# 1. Introduction

Magnetic wire-like structures with high aspect ratios, a link between nanoscale objects and the macroscale world, play important roles in both fundamental research and the development of modern materials [1,2]. The high aspect ratio endows the material with anisotropic properties and ensures that magnetization prefers to align with the long axis of the wires [3–5]. A longitudinal magnetic anisotropy has been widely reported to be related to strong shape anisotropy because the coercive fields in the wire direction are lower than those perpendicular to it [4,6–8]. Anisotropic magnetic wires also have higher magnetic moments than their spherical counterparts and therefore open up new possibilities for applications [5,9]. As a consequence, magnetic wire-like structures are of great interest in the development of new-generation spintronic devices [3], sensors [10], data storage technologies [11–13], biological and harsh environment applications [7,14], as well as many other potential applications [1,5,15]. To exploit their collective properties and the various applications in functional devices, many methods have been developed to produce wire-like structures [1,16].

The approaches used to obtain high aspect ratio magnetic wires can be generally classified into two main categories, i.e. direct synthesis and assembly methods [5]. The direct synthesis method includes solid-state, vapour-phase and liquid-/solution-based technique [1]. The solid-state and vapour-phase techniques are commonly costly and require complicated equipment [17,18]. The liquid-/solution-based synthesis technique, also known as a wet chemical process, is a prospective method in terms of applications with relatively low cost, simple equipment and high yield [1,19–21]. However, the morphology of particles, including the aspect ratio, always needs appropriate templates [22–27] or via chemical processes by tuning synthetic conditions, such as temperature, reaction time and reaction media, as well as the ratio and concentration of reagents [28,29]. The template-based method associates with its high cost and time-consuming nature, this method still cannot be adopted for large-scale production [1,30]. Although chemical process method is relatively low-cost and simple, some reaction parameters, including temperature and pH, as well as an external magnetic field (EMF), must be strictly controlled [8,18,31–33].

The assembly method commonly involves the cooperative combination of the EMF and a polymer, responsible for the self-assembly process of magnetic nanoparticles (MNPs) and gluing together the aligned nanoparticles, respectively [5]. EMF-assisted self-assembly of wire structures from MNPs is a simple, low-cost and environmentally friendly method, feasible for application in large-scale production [1,19,34–37]. The self-assembly is directed by an EMF via controlling the magnetic dipole–dipole interaction between MNPs [37–41]. The assembly is commonly performed in a solvent or during a solvent evaporation process to ensure that the MNPs have enough mobility to assemble [15]. A polymer is commonly employed to act as a linker for stabilizing the building blocks and retain the MNPs in the wire-like shape [42]. The combination of an EMF-assisted alignment and a polymer in solution can produce dense wire arrays after solvent evaporation [43]. Owing to the polymer stabilization and the strong interaction between MNPs, the assembled arrays and/or wire-like structures can remain stable when the EMF is removed [2]. The property and morphology of wire-like structures are influenced by several factors, including the EMF, the composition of the solvent and polymers, the pH of solutions and the reaction temperature. Among these, the EMF strength is the most crucial parameter for the formation of wires [1]. However, there are rare studies on the EMF-assisted wires from MNPs directly, especially under strong EMF. Our group succeeded in the rough alignment of core–shell $\alpha''$-$Fe_{16}N_2$ and $Fe_3O_4$ MNPs in the form of fibres and films, respectively, via magneto-electrospray under a 0.1 T EMF [44,45].

With the aim of fabricating magnetic wires in highly dense arrays, this study reports the magnetic wires directly assembled under 15 T EMF from the well-dispersed core–shell $\alpha''$-$Fe_{16}N_2$/$Al_2O_3$ and cube-like $Fe_3O_4$ MNPs with high and low magnetic isotropies, respectively, as well as the core–shell spindle-shaped $\alpha''$-$Fe_{16}N_2$/$Al_2O_3$ MNPs. We tried to investigate the shape factor of $\alpha''$-$Fe_{16}N_2$/$Al_2O_3$ NPs and develop the highly orientated $\alpha''$-$Fe_{16}N_2$/$Al_2O_3$ NPs assembly by applying the high magnetic field to enhance the magnetic performance by comparing with that of dispersed MNP without assembly. Magnetic properties and morphological characteristics of the obtained dense arrays were evaluated in detail and discussed involving the assembly mechanism and the effect of shape anisotropy.

# 2. Material and methods

## 2.1. Materials

The core–shell $\alpha''$-$Fe_{16}N_2$/$Al_2O_3$ and cube-like $Fe_3O_4$ MNPs with average diameters of 70 and 65 nm were used as the raw materials for the fabrication of magnetic wires in this study. The core–shell

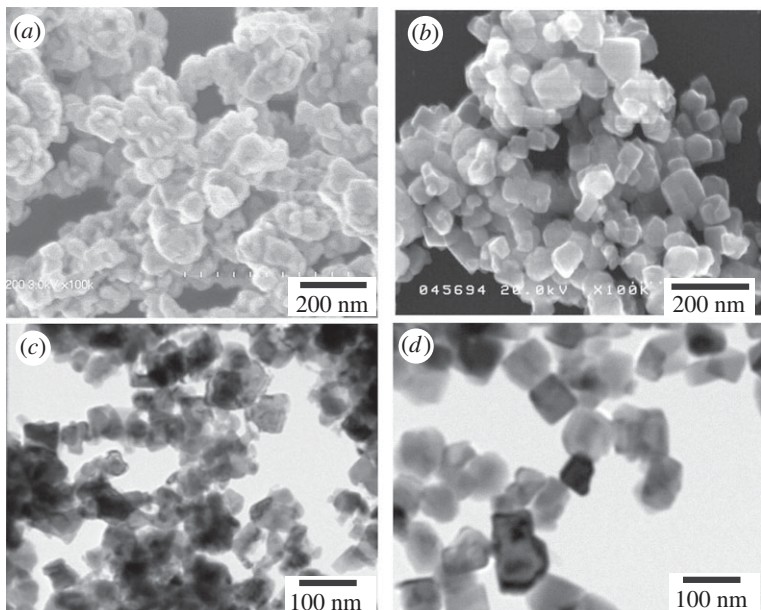

**Figure 1.** SEM images of core–shell $\alpha''$-Fe$_{16}$N$_2$/Al$_2$O$_3$ and cube-like Fe$_3$O$_4$ MNPs: (a) $\alpha''$-Fe$_{16}$N$_2$/Al$_2$O$_3$ and (b) Fe$_3$O$_4$ MNPs before dispersion; (c) $\alpha''$-Fe$_{16}$N$_2$/Al$_2$O$_3$ and (d) Fe$_3$O$_4$ MNPs after dispersion.

$\alpha''$-Fe$_{16}$N$_2$/Al$_2$O$_3$ MNPs with a shell thickness of 4.8 nm were prepared from cube-like Fe$_3$O$_4$ MNPs by the surface coating of Al$_2$O$_3$, reduction with H$_2$ gas and then nitridation with NH$_3$ gas, as detailed elsewhere [46]. The Fe$_3$O$_4$ MNPs with cube-like structures were prepared via a large-scale liquid precipitation method (US Patent no. 5843610, Toda Kogyo Co. Ltd, Japan) as described in our previous report [47]. An epoxy resin (3,4-epoxycyclohexylmethyl 3,4-epoxycyclohexanecarboxylate) was used as an MNP binder.

The colloidal dispersion of MNPs was prepared by using the bead-mill dispersion apparatus (dual axial type bead-mill, Kotobuki Industries, Co. Ltd, Japan), as described in our previous papers [48,49]. The precursor solutions for the wire fabrication were made from the mixture of epoxy resin (1.0 wt%) and MNPs (1.5 wt%) suspended in toluene with NH$_4$OH added to adjust the pH to approximately 9. Scanning electron microscope (SEM) images for core–shell $\alpha''$-Fe$_{16}$N$_2$/Al$_2$O$_3$ and Fe$_3$O$_4$ and MNPs before and after dispersion are shown in figure 1. Although the dispersion process was well controlled, sizes of $\alpha''$-Fe$_{16}$N$_2$/Al$_2$O$_3$ MNPs were still decreased after dispersion owing to their strong aggregation effect. Average diameters after dispersion are 45.3 and 64.7 nm for $\alpha''$-Fe$_{16}$N$_2$/Al$_2$O$_3$ and Fe$_3$O$_4$MNPs, respectively. To determine the implications of the results obtained in this study, spindle-shaped core–shell $\alpha''$-Fe$_{16}$N$_2$/Al$_2$O$_3$ MNPs 110 nm in length and 18 nm in width (detailed characteristics were reported in our previous paper [46]) were also used to fabricate wires to reveal the effect of shape anisotropy.

## 2.2. Fabrication technique

The precursor solutions with the MNP suspension were shaken in an ultrasonic bath (Sine Sonic UA-100, 36 kHz, 100 W) for 30 min before fabrication. The solutions with a volume of 30 ml were then transferred into quartz beakers with inner diameters of 35 mm and inner heights of 15 mm. The beaker was fixed to another quartz beaker, with an inner diameter of 45 mm, a sieve mesh bottom and a hood. The hood was connected to a flexible tube for pumping nitrogen gas with a constant flow rate of 1 l min$^{-1}$, which was controlled by a mass flow meter. The input of nitrogen gas assisted the evaporation of the solvent in the precursor solutions and flowed out from the bottom of the outside beaker. The packed beaker system, including the solutions, was placed at the centre position of a superconducting magnet (Oxford Instruments, Spectromag-1000) through a bore tube (50 mm). A holder was placed at the bottom of the outside breaker to fix the packed system, as shown in figure 2a. The maximum strength of the EMF was 15.0 T in the vertical direction at the centre position. A detailed description of the superconducting magnet system has been reported elsewhere with the magnetic field spatial distribution along the centre axis shown in figure 2b [50]. The solutions were maintained in the strong

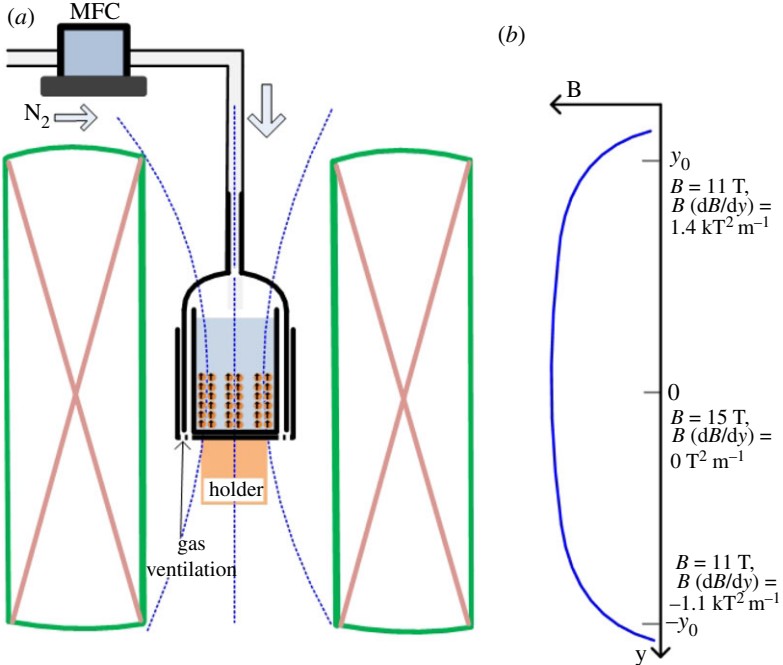

**Figure 2.** (*a*) Cross-section of the magnet and schematic illustration of the fabrication process of magnetic wire arrays from MNPs under an EMF of 15 T. (*b*) Spatial distribution of the magnetic flux density along the vertical axis in the centre of the magnet.

EMF for 24 h evaporation. The whole fabrication process was operated at room temperature, approximately 25°C. Obtained dried wire arrays were scratched for further characterization. Another fabrication via the same method for spindle-shaped core–shell $\alpha''$-$Fe_{16}N_2$/$Al_2O_3$ MNPs was performed under 0.8 T. The 0.8 T was created at the surface centre of a permanent magnet, which was the only dependable and available one in the laboratory.

## 2.3. Characterization

The morphologies of the MNPs and the fabricated arrays were observed using an SEM (Hitachi S-5000, Japan). Their crystalline structures were examined using X-ray diffraction (XRD; D2 Phaser, Bruker, Germany), while assigned Miller indices of the peaks were obtained from the JCPDS database. Their magnetic properties were evaluated using a superconducting quantum interference device (SQUID, Quantum Design, Tokyo, Japan), which was operated at 300 K. Magnetization was measured as a function of the applied field from 0 to 50 kOe.

# 3. Results

## 3.1. Morphology and crystalline structure

Figure 3 presents typical SEM images of the fabricated arrays assembled from core–shell $\alpha''$-$Fe_{16}N_2$/$Al_2O_3$ (figure 3*a–d*) and cube-like $Fe_3O_4$ MNPs (figure 3*e,f*) under the EMF of 15.0 T. The highly aligned wires are composed of MNPs, as shown in the gradual magnification of wire SEM images from (figure 3*a–d*). The well-dispersed single-domain sized $\alpha''$-$Fe_{16}N_2$/$Al_2O_3$ MNPs participated in the formation of wires along the EMF direction (figure 1*c*). Different from previous nanowires assembled through dipolar interactions between single-domain MNPs and with the same diameter of single-domain MNPs [35,51,52], the assembled wires in this study were not single MNP chains and contained many MNPs in the cross-section of every single wire. Thus, the average diameter of the wires was larger than that of the composite MNPs. The diameters of wires, of the order of 1 µm, are difficult to estimate because they are overlapped with each other. The average length of the assembled wires is estimated to be approximately 10 mm. Therefore, the ratio between length and width is of the order of $10^4$. The microscope images show that the assembled wires were nearly straight and parallel to each other.

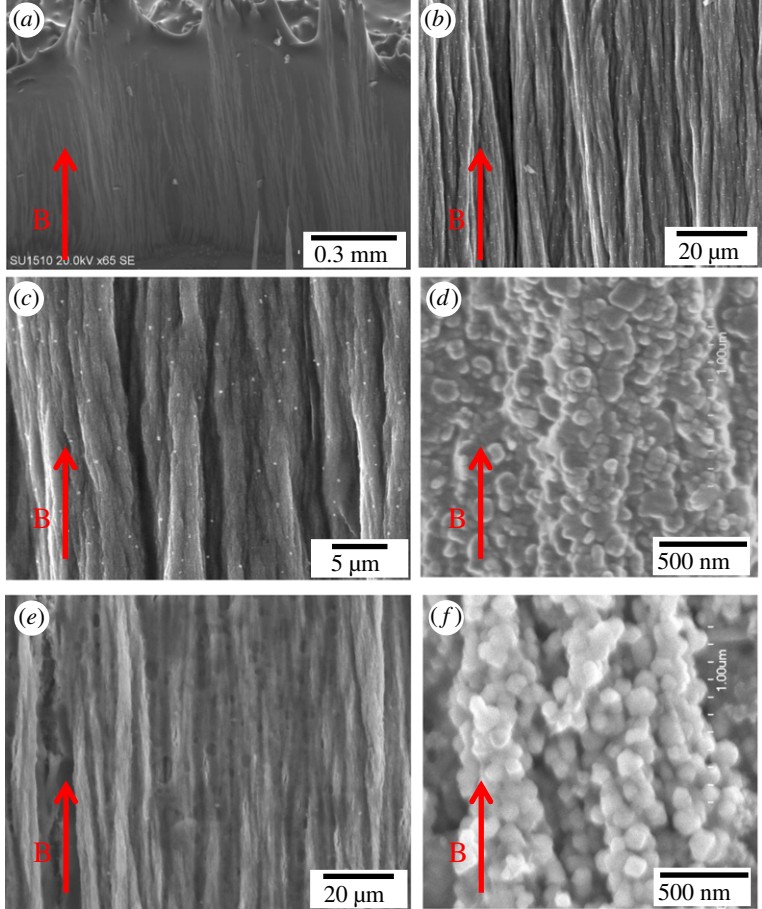

**Figure 3.** SEM images with different magnifications of fabricated arrays assembled from ($a$–$d$) core–shell $\alpha''$-Fe$_{16}$N$_2$/Al$_2$O$_3$ MNPs and ($e,f$) Fe$_3$O$_4$ MNPs under an EMF of 15 T.

Figure 4 shows the XRD patterns of dispersed MNPs, featuring $\alpha''$-Fe$_{16}$N$_2$/Al$_2$O$_3$ and Fe$_3$O$_4$ crystalline structures, and those of their assembled arrays under an EMF of 15.0 T. Different from MNPs, the XRD patterns of the fabricated arrays possess a slightly uphill baseline and obvious noise on baselines. This difference probably originates from the addition of the epoxy resin during the production of arrays. Similar phenomena were also observed for fabricating $\alpha''$-Fe$_{16}$N$_2$ contained fibre sand carbon nanotube films in our previous studies [44,53,54]. The XRD peaks of (202), (220), (004) and (400) from crystalline $\alpha''$-Fe$_{16}$N$_2$/Al$_2$O$_3$ MNPs are also visible for their assembled arrays in the $2\theta$ range of 25°–70°, while peaks of (111), (311), (222), (400), (422), (511) and (400) from single-crystalline Fe$_3$O$_4$ MNPs (as reported in our previous paper [47]) are shown for their assembled arrays in the same $2\theta$ range. The patterns indicate that the fabricated arrays retained the inherent crystalline properties of their composite MNPs.

For the highly magnetic isotropic $\alpha''$-Fe$_{16}$N$_2$ MNPs and their arrays, the (004) peak has a vertical direction along the $c$-axis direction, while the (220) peak is in the horizontal direction [54]. When the EMF was applied along the array direction, the (004) and (220) diffraction peaks increased and decreased in the array direction, respectively. The XRD patterns were obtained in the parallel direction of the arrays. By contrast, the (004) and (220) diffraction peaks decreased and increased in the vertical direction of the arrays, respectively, as shown in figure 4. This result is consistent with our previous report on $\alpha''$-Fe$_{16}$N$_2$/Al$_2$O$_3$ films synthesized under various EMF conditions [54]. Therefore, the XRD patterns suggest that the strong EMF leads to increasing the alignment of the $c$-axis of the MNPs in the array direction.

For the Fe$_3$O$_4$ MNPs with low magnetic anisotropy, the (311) plane is in the vertical direction of the $c$-axis [55]. Similar to the effect of the EMF on the XRD patterns of $\alpha''$-Fe$_{16}$N$_2$/Al$_2$O$_3$ wires, the (311) peak decreased in the vertical direction of arrays, as shown in figure 4. This observation further identifies the effect of the EMF on the alignment of the $c$-axis of the MNPs in the array direction.

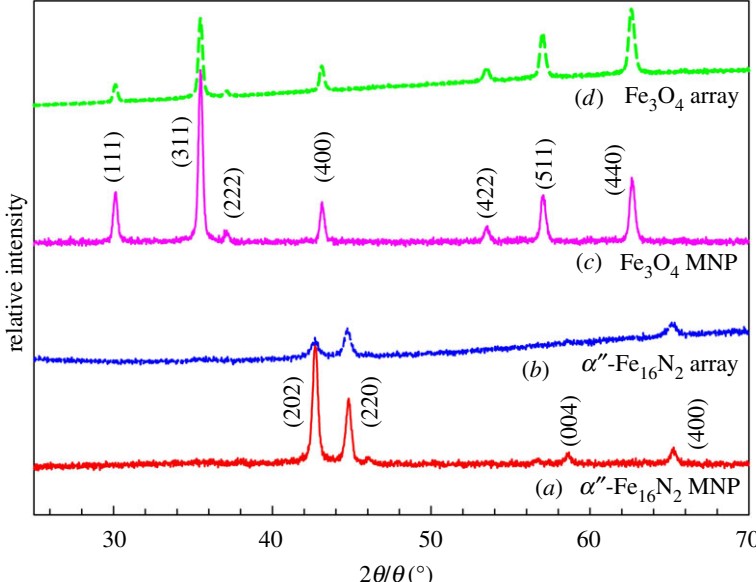

**Figure 4.** XRD patterns of (*a*) core–shell α″-Fe₁₆N₂/Al₂O₃ MNPs and (*b*) their assembled arrays, (*c*) Fe₃O₄ MNPs and (*d*) their assembled arrays.

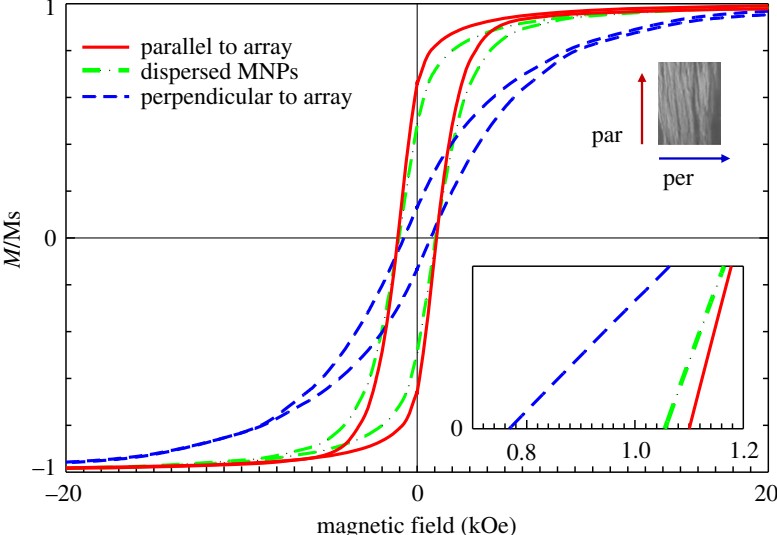

**Figure 5.** Magnetic characterization of dispersed core–shell α″-Fe₁₆N₂ /Al₂O₃ MNPs and their assembled array under an EMF of 15 T with the measured magnetic field applied parallel and perpendicular to arrays.

## 3.2. Magnetic property of arrays

Figure 5 shows the magnetic hysteresis (*M–H*) loops of the dispersed α″-Fe₁₆N₂/Al₂O₃ MNPs and aligned arrays with the measured magnetic field parallel and perpendicular to the wire direction. These loops were obtained by SQUID measurements at 300 K. The magnetization (*M*) of each *M–H* loop is normalized by its corresponding saturation magnetization (Ms). The Ms for the fabricated array including the fabricated epoxy resin was 181 emu g⁻¹. Compared with the dispersed α″-Fe₁₆N₂ MNPs, the shape of the *M–H* loop for the fabricated α″-Fe₁₆N₂ array appeared to be more rectangular and spindle-shaped when the measured field was applied parallel and perpendicular to the array direction, respectively, as clearly shown in figure 5. The ratios of remanence (Mr) values and Ms were 49%, 66% and 13% for the dispersed α″-Fe₁₆N₂ MNPs, arrays in the parallel and arrays in the perpendicular directions, respectively. The magnetic coercivity (Hc) values were 1.053, 1.120 and 0.766 kOe for the three cases, respectively. The characteristics of the *M–H* loops suggest that the

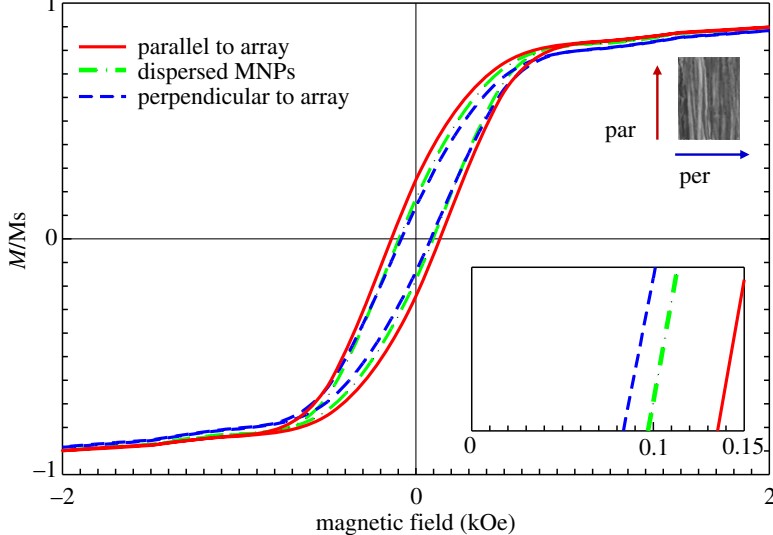

**Figure 6.** Magnetic characterization of dispersed $Fe_3O_4$ MNPs and their assembled array under an EMF of 15 T with the measured magnetic field applied parallel and perpendicular to arrays.

**Table 1.** Summary of magnetic properties at 300 K for dispersed $\alpha''$-$Fe_{16}N_2$/$Al_2O_3$ and $Fe_3O_4$ MNPs and their assembled arrays under an EMF of 15 T with the measured magnetic field applied parallel and perpendicular to arrays.

| array | EMF direction | Mr/Ms (%) | Hc (kOe) |
|---|---|---|---|
| $Fe_{16}N_2$ | particles | 49 | 1.05 |
| | parallel | 66 | 1.12 |
| | perpendicular | 13 | 0.77 |
| $Fe_3O_4$ | particles | 19 | 0.097 |
| | parallel | 26 | 0.136 |
| | perpendicular | 14 | 0.083 |

fabricated $\alpha''$-$Fe_{16}N_2$ wires exhibit enhanced magnetic properties along the wire direction with a larger Ms, Mr/Ms and Hc values than those of the dispersed MNPs and those in the vertical direction of the wires. The anisotropy field, calculated from the measured easy axis (parallel to arrays) and hard axis (perpendicular to arrays) loops of the reduced magnetization [56], is 6.235 kOe for the $\alpha''$-$Fe_{16}N_2$/ $Al_2O_3$ arrays. Although the increase in Mr/Ms and Hc is only 34.7% and 6.4% for the dispersed $\alpha''$-$Fe_{16}N_2$/$Al_2O_3$ MNPs, respectively, the result identifies that the used fabrication method is valid in producing arrays and enhancing their magnetic properties.

Figure 6 shows similar magnetic properties of fabricated $Fe_3O_4$ arrays to that of the $\alpha''$-$Fe_{16}N_2$/$Al_2O_3$ arrays, although the $Fe_3O_4$ MNPs show less magnetic anisotropy. The Ms value for the fabricated $Fe_3O_4$ array was 65 emu $g^{-1}$. The ratios of Mr/Ms were 19%, 26% and 14% for the dispersed $Fe_3O_4$ MNPs, arrays in the parallel and perpendicular directions, respectively, while the Hc values were approximately 0.097, 0.136 and 0.083 kOe for the three cases. The anisotropy field for the $Fe_3O_4$ arrays is 0.202 kOe, which is much lower than that of the $\alpha''$-$Fe_{16}N_2$/$Al_2O_3$ arrays. Thus, there is less difference for the hysteresis loops with the magnetic field applied in the directions perpendicular and parallel to the $Fe_3O_4$ arrays than that to the $\alpha''$-$Fe_{16}N_2$/$Al_2O_3$ arrays, as well as less coupled NPs in figure 3f. The characteristics of $Fe_3O_4$ M–H loops were consistent with that of the $\alpha''$-$Fe_{16}N_2$ M–H loops. This further identifies that the magnetic properties of the fabricated wires were enhanced by the EMF of 15 T. However, compared with the enhancement for the $Fe_3O_4$ array, the enhancement for the $\alpha''$-$Fe_{16}N_2$/$Al_2O_3$ arrays was much larger, owing to the higher isotropic of $\alpha''$-$Fe_{16}N_2$/$Al_2O_3$ MNPs than that of $Fe_3O_4$ MNPs. Magnetic properties at 300 K for dispersed $\alpha''$-$Fe_{16}N_2$/$Al_2O_3$ and $Fe_3O_4$ MNPs and their assembled arrays are summarized in table 1.

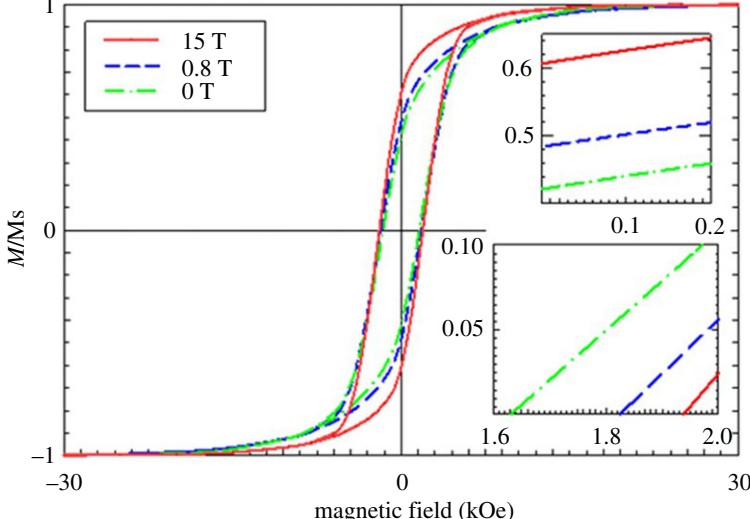

**Figure 7.** Magnetic characterization of dispersed spindle $\alpha''$-$Fe_{16}N_2$/$Al_2O_3$ MNPs and their assembled arrays under EMFs of 0.8 and 15 T with the measured magnetic field applied parallel to arrays.

Magnetic properties at 300 K for dispersed $\alpha''$-$Fe_{16}N_2$/$Al_2O_3$ and $Fe_3O_4$ MNPs and their assembled arrays are summarized in table 1. The application of a strong EMF with an intensity of 15 T has been demonstrated to be an effective approach to fabricate magnetic arrays from typical nanoparticles with high and low magnetic anisotropy, i.e. $\alpha''$-$Fe_{16}N_2$/$Al_2O_3$ and $Fe_3O_4$ MNPs.

The induced magnetic field strength of the arrays fabricated from $\alpha''$-$Fe_{16}N_2$/$Al_2O_3$ and $Fe_3O_4$ MNPs was directly measured using a magnetometer (TM-801, Kanetec, Japan). The magnetic values under the bottom of inner beakers used for fabrication were approximately 0.2 T and higher than the detection lower limit (less than 0.1 T). The generated detectable magnetic field suggests that the fabricated arrays from isotropic MNPs possess the potential application in tiny magnets [57].

## 3.3. Arrays fabricated from spindle-shaped MNPs

The fabricated wires possess a shape anisotropy and cause the difference of the magnetic performance in the parallel and perpendicular directions of the wires. The enhancement of magnetic properties for the fabricated straight wires in the parallel direction of the wires is due to the increased magnetic dipole–dipole interaction between MNPs in the wire direction [37]. In a new scenario, if elongated MNPs with shape anisotropy can rotate freely in a solvent, an EMF can make them orient along a magnetic easy axis oriented to the magnetic line and inevitably induce the formation of arrays. Thus, the shape anisotropy of MNPs can increase the alignment of single-domain MNPs with magnetic easy axes under an EMF [37,58]. The shape anisotropy will be enhanced if the external magnetic field is applied to spindle-shape MNPs.

Figure 7 presents the comparison of the magnetic property for the dispersed spindle-shape core–shell $\alpha''$-$Fe_{16}N_2$/$Al_2O_3$ MNPs and their aligned arrays under EMFs of 0.8 and 15 T with the magnetic field, as measured by SQUID, parallel to the wire direction. The magnetic properties of the spindle-shaped $\alpha''$-$Fe_{16}N_2$/$Al_2O_3$ MNPs 110 nm in length and 18 nm in width have been described in detail elsewhere [46]. The ratio of Mr/Ms and Hc increased with the EMF strength, as summarized in table 2. The increase in EMF strength caused the increase in the alignment of MNPs in arrays and resulted in an enhancement of magnetic performance of the fabricated arrays. This result is consistent with our previous report on the alignment of $\alpha''$-$Fe_{16}N_2$/$Al_2O_3$ MNPs in polymer films [54], as well as other investigations on the EMF intensity effect on wire-like structures via the direct synthesis under a weak EMF (less than 0.5 T) [33,51,59–61] and strong EMF (less than 1.4 T; 0–10 T) [8].

The ratio of Mr/Ms increased 15.0% and 44.4% under 0.8 and 15 T, respectively, while the Hc increased 11.7% and 19.0% for the two cases, respectively. The increase in Mr/Ms and Hc for the spindle-shaped $\alpha''$-$Fe_{16}N_2$/$Al_2O_3$ are both higher than those for the cube-like one (34.7% and 6.4%) under the same EMF of 15 T. The increased enhancement of magnetic properties is due to the spindle-shape anisotropy, which increases the alignment of single-domain MNPs along the EMF direction.

**Table 2.** Summary of magnetic properties at 300 K for the arrays assembled from spindle-shape $\alpha''$-Fe$_{16}$N$_2$/Al$_2$O$_3$ MNPs under EMF of 0.8 and 15 T with the measured magnetic field applied parallel to the arrays.

| EMF intensity (T) | Mr/Ms (%) | Hc (kOe) |
| --- | --- | --- |
| 0 | 42.1 | 1.63 |
| 0.8 | 48.4 | 1.82 |
| 15 | 60.8 | 1.94 |

# 4. Conclusion

Magnetic arrays composed of highly dense wires have been successfully fabricated from well-dispersed cube-like Fe$_3$O$_4$ and core–shell cube/spindle-shaped $\alpha''$-Fe$_{16}$N$_2$/Al$_2$O$_3$ MNPs under an EMF of 15 T. The length and the aspect ratio of the fabricated arrays are of the order of 1 mm and 1000, respectively. The fabricated arrays possess enhanced magnetic properties along the wire direction compared with the dispersed MNPs. The comparison study of $M$–$H$ loops reveals that the Mr/Ms and Hc values along the wire direction are both significantly higher than those of the MNPs. Compared with the cube-like $\alpha''$-Fe$_{16}$N$_2$/Al$_2$O$_3$ MNPs, the spindle-shaped ones have the advantage in the enhancement of magnetic performance of their fabricated arrays owing to the shape anisotropy. Compared with the Fe$_3$O$_4$ MNPs with low magnetic anisotropy, the assembled arrays from highly isotropic $\alpha''$-Fe$_{16}$N$_2$/Al$_2$O$_3$ MNPs generate a detectable magnetic field of approximately 0.2 T. These results indicate that the developed technique is feasible to fabricate magnetic wires in highly dense arrays at a large scale for various potential applications.

Data accessibility. The data for XRD and magnetic hysteresis loops are provided as electronic supplementary material.
Authors' contributions. Q.L., T.I. and K.O. designed the research. K.O. and T.O. supervised the research. Q.L., C.W.K. and T.I. performed the experiments. Q.L. and C.W.K. contributed new reagents/analytic tools. Q.L., C.W.K. and K.O., with inputs from other co-authors, performed data analysis and wrote the paper.
Competing interests. The authors declare that there are no competing interests involved.
Funding. This work was supported by JSPS (Japan Society for the Promotion of Science) KAKENHI (grant nos. 26709061 and 16K13642). This work was partly supported by the Center for Functional Nano Oxide at Hiroshima University (Japan).
Acknowledgements. The authors thank Prof. Yoshihisa Fujiwara and Associate Prof. Takahiro Onimaru of Hiroshima University for the assistance in operating the superconducting magnet and performing the SQUID measurements.

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
