## [Reviewer comments · Royal Society Open Science]

Review History

RSOS-190534.R0 (Original submission)

Review form: Reviewer 1

Is the manuscript scientifically sound in its present form?

No

Are the interpretations and conclusions justified by the results?

No

Is the language acceptable?

Yes

Is it clear how to access all supporting data?

Not applicable

Have you any concerns about statistical analyses in this paper?

No

Recommendation?

Major revision is needed (please make suggestions in comments)

Comments to the Author(s)

According to the authors, the manuscript describes a method to produce dense arrays of nanowires (NWs) with magnetic coercivity and remanent magnetization "much higher than that of the nanoparticles (NPs)". However, this is not supported by reported data. In figure 5, hysteresis loops of array and NPs are very similar, with only a very small increase of coercivity (less than 6.5 %). The case of figure 6 is even worse: the hysteresis loops measured in to perpendicular directions of the arrays and the one measured in the NPs are essentially the same, showing no contribution from shape anisotropy in these NWs. The third figure related magnetism (figure 7), shows no change in coercivity and small increase of remanence upon applied of high magnetic field. It is quite surprising for me that the application of 0.8 T (quite a large magnetic field) produces almost no effect in the system.

It is also remarkable that most of the hysteresis loops are measured only with the applied field parallel to the axis of the nanowires and, therefore, magnetization processes are mainly due to domain wall nucleation, pinning and movement. Under these conditions it is difficult to evaluate the shape anisotropy induced in the nanowires. It would be important to show the hysteresis loops measured with the applied field perpendicular to the wire axis. Under these conditions it is easier to measure the anisotropy field.

Magnetic properties of nanowires are governed by shape anisotropy and many examples can be found in literature. In the case of the reported nanowires, I cannot see the expected behavior for magnetic nanowires.

In addition to these remarks, there are other issues that should be addressed.

It is true that, from the XRD, it seems that the material is better oriented because there are less reflections in the diffractograms. However, the intensity of the peaks is lower and, in the particular case of the Fe₁₆N₂ array, the signal/noise ratio very poor. It seems that, when particles are oriented to form the nanowires, their crystal quality is worse. Is there any reason for these?

Figure 3 shows SEM images of the NWs. From the figures (in particular panels b, c and e), some texture is shown. However, when zooming (panels d and f), no evidence of NWs formation is clear, but clusters or disordered particles.

To sum up, I believe the method proposed by the authors could be interesting, but more clear experimental proofs of the formation of nanowires from the structural and magnetic point of view should be presented.

Review form: Reviewer 2

Is the manuscript scientifically sound in its present form?

Yes

Are the interpretations and conclusions justified by the results?

No

Is the language acceptable?

No

Is it clear how to access all supporting data?

Not Applicable

Do you have any ethical concerns with this paper?

No

Have you any concerns about statistical analyses in this paper?

I do not feel qualified to assess the statistics

Recommendation?

Reject

Comments to the Author(s)

The use of magnetic field as a parameter, which allows a self-assembly process of magnetic nanoparticles, constitutes very interesting topic in the material science. In particular, this approach plays an important role in the synthesis of wire-like nanomaterials composed of nanoparticles.

Even if the reviewed manuscript presents the quite interesting results, I do not recommend this work to be published in the Royal Society Open Science in the present version due to some factors. Few of them are listed below.

- 1) The reviewed manuscript contains a lot of grammar errors and, in general, it is written using poor English. Therefore, the authors should improve all language mistakes before the new submission.
- 2) The presented experimental results are not entirely convincing. There is no explanation why the authors decided to apply the external magnetic field (EMF) of 0, 0.8 and 15 T for studying the alignment of spindle-shape core-shell α "-Fe₁₆N₂/Al₂O₃ MNPs?
- 3) At the beginning, the authors present the dimensions of initial materials taken to the dispersion process. Then, they claim that this process leads to a partial damaging of α "-Fe₁₆N₂/Al₂O₃ MNPs (comparison shown in Fig. 1) and their average dimensions become the same. How is it possible?
- 4) The explanations provided on pages 10-11, why the intensities of XRD peaks vary, are unclear. They need to be improved.
- 5) No information is provided which the XRD database has been used in order to assign the peak positions to the Miller indices.
- 6) Also, the explanation provided on page 14, why the isotropic MNPs could be used for sensors, is not clear enough. This part should contain more information.

Decision letter (RSOS-190534.R0)

22-Jul-2019

Dear Dr Ogi:

Manuscript ID: RSOS-190534

Title: "Enhanced magnetic performance of aligned wires assembled from nanoparticles: from nanoscale to macroscale"

Thank you for submitting the above manuscript to Royal Society Open Science. Your paper was sent to reviewers and their comments are included at the bottom of this letter. I apologise that this has taken longer than usual.

In view of the concerns raised by the reviewers, the manuscript has been rejected in its current form. However, a new manuscript may be submitted which takes into consideration these comments.

Please note that resubmitting your manuscript does not guarantee eventual acceptance, and that your resubmission will be subject to peer review before a decision is made.

Your resubmitted manuscript should be submitted by 19-Jan-2020. If you are unable to submit by this date please contact the Editorial Office.

On behalf of the Subject Editor Professor Anthony Stace and the Associate Editor Professor Tobias Hertel

REVIEWER(S) REPORTS:
Associate Editor Comments to Author ():
RSC Associate Editor:
Comments to the Author:
(There are no comments.)

RSC Subject Editor:
Comments to the Author:
(There are no comments.)

Reviewers' Comments to Author:
Reviewer: 1

Comments to the Author(s)
According to the authors, the manuscript describes a method to produce dense arrays of nanowires (NWs) with magnetic coercivity and remanent magnetization "much higher than that of the nanoparticles (NPs)". However, this is not supported by reported data. In figure 5, hysteresis loops of array and NPs are very similar, with only a very small increase of coercivity (less than 6.5 %). The case of figure 6 is even worse: the hysteresis loops measured in to perpendicular directions of the arrays and the one measured in the NPs are essentially the same,

showing no contribution from shape anisotropy in these NWs. The third figure related magnetism (figure 7), shows no change in coercivity and small increase of remanence upon applied of high magnetic field. It is quite surprising for me that the application of 0.8 T (quite a large magnetic field) produces almost no effect in the system.

It is also remarkable that most of the hysteresis loops are measured only with the applied field parallel to the axis of the nanowires and, therefore, magnetization processes are mainly due to domain wall nucleation, pinning and movement. Under these conditions it is difficult to evaluate the shape anisotropy induced in the nanowires. It would be important to show the hysteresis loops measured with the applied field perpendicular to the wire axis. Under these conditions it is easier to measure the anisotropy field.

Magnetic properties of nanowires are governed by shape anisotropy and many examples can be found in literature. In the case of the reported nanowires, I cannot see the expected behavior for magnetic nanowires.

In addition to these remarks, there are other issues that should be addressed.

It is true that, from the XRD, it seems that the material is better oriented because there are less reflections in the diffractograms. However, the intensity of the peaks is lower and, in the particular case of the Fe₁₆N₂ array, the signal/noise ratio very poor. It seems that, when particles are oriented to form the nanowires, their crystal quality is worse. Is there any reason for these?

Figure 3 shows SEM images of the NWs. From the figures (in particular panels b, c and e), some texture is shown. However, when zooming (panels d and f), no evidence of NWs formation is clear, but clusters or disordered particles.

To sum up, I believe the method proposed by the authors could be interesting, but more clear experimental proofs of the formation of nanowires from the structural and magnetic point of view should be presented.

Reviewer: 2

Comments to the Author(s)

The use of magnetic field as a parameter, which allows a self-assembly process of magnetic nanoparticles, constitutes very interesting topic in the material science. In particular, this approach plays an important role in the synthesis of wire-like nanomaterials composed of nanoparticles.

Even if the reviewed manuscript presents the quite interesting results, I do not recommend this work to be published in the Royal Society Open Science in the present version due to some factors. Few of them are listed below.

- 1) The reviewed manuscript contains a lot of grammar errors and, in general, it is written using poor English. Therefore, the authors should improve all language mistakes before the new submission.
- 2) The presented experimental results are not entirely convincing. There is no explanation why the authors decided to apply the external magnetic field (EMF) of 0, 0.8 and 15 T for studying the alignment of spindle-shape core-shell α -Fe₁₆N₂/Al₂O₃ MNPs?
- 3) At the beginning, the authors present the dimensions of initial materials taken to the dispersion process. Then, they claim that this process leads to a partial damaging of α -Fe₁₆N₂/Al₂O₃ MNPs (comparison shown in Fig. 1) and their average dimensions become the same. How is it possible?
- 4) The explanations provided on pages 10-11, why the intensities of XRD peaks vary, are unclear. They need to be improved.

- 5) No information is provided which the XRD database has been used in order to assign the peak positions to the Miller indices.
- 6) Also, the explanation provided on page 14, why the isotropic MNPs could be used for sensors, is not clear enough. This part should contain more information.

Author's Response to Decision Letter for (RSOS-190534.R0)

See Appendix A.

RSOS-191656.R0

Review form: Reviewer 1

Is the manuscript scientifically sound in its present form?

No

Are the interpretations and conclusions justified by the results?

No

Is the language acceptable?

Yes

Do you have any ethical concerns with this paper?

No

Have you any concerns about statistical analyses in this paper?

No

Recommendation?

Major revision is needed (please make suggestions in comments)

Comments to the Author(s)

After the revision, the manuscript has been improved, in particular the part related to the formation of nanowires and the study of the structure. However, the analysis of the magnetic properties is very weak and the authors have not fully addressed the previous concerns.

In particular, I suggested measuring with magnetic field applied in the direction perpendicular to the wires to determine the anisotropy field. This parameter is important to determine whether the magnetic properties come from the wire structure or not. This parameter has not been calculated.

I agree that figure 5 reflects, probably, the shape anisotropy in the magnetic properties due to the wire formation, although it is not very clear in the text. However, the authors do not explain why, in figure 6, all hysteresis loops are basically identical, with very small variation. This, in fact, is an indication of the existence of decoupled particles and not a wire. This, in fact, is also shown in figure 3f: there is no wires but chains of decoupled NPs.

Authors talk about magnetic isotropy in several parts of the manuscript. Do they refer to magnetic anisotropy?

Finally, with the reported values of M_r/M_s and H_c , I cannot see how this material can be used as permanent magnets.

So, to conclude, as mentioned before, I believe the manuscript has potential interest. However, I believe authors should center the manuscript in the fabrication and structural characterization, and rewrite the magnetic properties to support the formation of real wires and not agglomeration of particles.

Review form: Reviewer 2

Is the manuscript scientifically sound in its present form?

Yes

Are the interpretations and conclusions justified by the results?

Yes

Is the language acceptable?

Yes

Do you have any ethical concerns with this paper?

No

Have you any concerns about statistical analyses in this paper?

No

Recommendation?

Accept with minor revision (please list in comments)

Comments to the Author(s)

I was asked again to review the re-submitted version of manuscript entitled 'Enhanced magnetic performance of aligned wires assembled from nanoparticles: from nanoscale to macroscale'. My opinion about the topic of the performed investigations does not change. I still think that the use of magnetic field as a manufacturing parameter is very interesting approach to obtain the anisotropic magnetic wire-like materials.

The new version of manuscript is more clear than the previous one and the results are presented in more convincing manner. The previously mentioned lapses are well addressed. Therefore, I would like to recommend this work to be published in the Royal Society Open Science after minor revision. The list of remarks is included below.

1) Again, I need to point out the lack of explanation why the Authors have decided to apply such a high external magnetic field (15 T) in order to obtain the wires of α -Fe₁₆N₂/Al₂O₃ MNPs. In general, they have presented that the much weaker magnetic field of 0.8 T is not sufficient for assembling highly-anisotropic material but between 0.8 T and 15 T is a high difference. Could the Authors comment this issue?

2) The second concern is associated with the sentence included in abstract. The Authors claim that "(...) This study reports a facile method to fabricate arrays of magnetic wires (...)". I agree that the concept of the manufacturing process is rather facile but the performance of this process might not be so easy due to the application of very high magnetic field of 15 T. Could the Authors comment this issue?

3) Could the Authors comment why the values of saturation magnetizations measured for the magnetic field applied parallel and perpendicular to the array of wires are different?

4) Also, I found three mistyping in the re-submitted manuscript.

- page 9 line 44; instead of 'weren't' should be 'were not'
- page 19 line 24; instead of 'Table 1' should be 'Table 2'
- page 20 line 10; missing unit right behind '1000'

Decision letter (RSOS-191656.R0)

19-Dec-2019

Dear Dr Ogi:

Title: Enhanced magnetic performance of aligned wires assembled from nanoparticles: from nanoscale to macroscale

Manuscript ID: RSOS-191656

The editor assigned to your paper has now received comments from reviewers. I apologise that this has taken longer than usual. We would like you to revise your paper in accordance with the referee and Subject Editor suggestions which can be found below (not including confidential reports to the Editor). Please note this decision does not guarantee eventual acceptance.

Please submit a copy of your revised paper before 11-Jan-2020. Please note that the revision deadline will expire at 00.00am on this date. If we do not hear from you within this time then it will be assumed that the paper has been withdrawn. In exceptional circumstances, extensions may be possible if agreed with the Editorial Office in advance. We do not allow multiple rounds of revision so we urge you to make every effort to fully address all of the comments at this stage. If deemed necessary by the Editors, your manuscript will be sent back to one or more of the original reviewers for assessment. If the original reviewers are not available we may invite new reviewers.

On behalf of the Subject Editor Professor Anthony Stace and the Associate Editor Professor Tobias Hertel.

RSC Associate Editor
 Comments to the Author:
 (There are no comments.)

Reviewers' Comments to Author:
 Reviewer: 2

Comments to the Author(s)

I was asked again to review the re-submitted version of manuscript entitled 'Enhanced magnetic performance of aligned wires assembled from nanoparticles: from nanoscale to macroscale'. My opinion about the topic of the performed investigations does not change. I still think that the use of magnetic field as a manufacturing parameter is very interesting approach to obtain the anisotropic magnetic wire-like materials.

The new version of manuscript is more clear than the previous one and the results are presented in more convincing manner. The previously mentioned lapses are well addressed. Therefore, I would like to recommend this work to be published in the Royal Society Open Science after minor revision. The list of remarks is included below.

- 1) Again, I need to point out the lack of explanation why the Authors have decided to apply such a high external magnetic field (15 T) in order to obtain the wires of α -Fe₁₆N₂/Al₂O₃ MNPs. In general, they have presented that the much weaker magnetic field of 0.8 T is not sufficient for assembling highly-anisotropic material but between 0.8 T and 15 T is a high difference. Could the Authors comment this issue?
- 2) The second concern is associated with the sentence included in abstract. The Authors claim that "(...) This study reports a facile method to fabricate arrays of magnetic wires (...)". I agree that the concept of the manufacturing process is rather facile but the performance of this process might not be so easy due to the application of very high magnetic field of 15 T. Could the Authors comment this issue?
- 3) Could the Authors comment why the values of saturation magnetizations measured for the magnetic field applied parallel and perpendicular to the array of wires are different?
- 4) Also, I found three mistyping in the re-submitted manuscript.
 - page 9 line 44; instead of 'weren't' should be 'were not'
 - page 19 line 24; instead of 'Table 1' should be 'Table 2'
 - page 20 line 10; missing unit right behind '1000'

Reviewer: 1

Comments to the Author(s)

After the revision, the manuscript has been improved, in particular the part related to the formation of nanowires and the study of the structure. However, the analysis of the magnetic properties is very weak and the authors have not fully addressed the previous concerns.

In particular, I suggested measuring with magnetic field applied in the direction perpendicular to the wires to determine the anisotropy field. This parameter is important to determine whether the magnetic properties come from the wire structure or not. This parameter has not been calculated.

I agree that figure 5 reflects, probably, the shape anisotropy in the magnetic properties due to the wire formation, although it is not very clear in the text. However, the authors do not explain why, in figure 6, all hysteresis loops are basically identical, with very small variation. This, in fact, is an indication of the existence of decoupled particles and not a wire. This, in fact, is also shown in figure 3f: there are no wires but chains of decoupled NPs.

Authors talk about magnetic isotropy in several parts of the manuscript. Do they refer to magnetic anisotropy?

Finally, with the reported values of M_r/M_s and H_c , I cannot see how this material can be used as permanent magnets.

So, to conclude, as mentioned before, I believe the manuscript has potential interest. However, I believe authors should center the manuscript in the fabrication and structural characterization, and rewrite the magnetic properties to support the formation of real wires and not agglomeration of particles.

Author's Response to Decision Letter for (RSOS-191656.R0)

See Appendix B.

Decision letter (RSOS-191656.R1)

26-Mar-2020

Dear Dr Ogi:

Title: Enhanced magnetic performance of aligned wires assembled from nanoparticles: from nanoscale to macroscale

Manuscript ID: RSOS-191656.R1

It is a pleasure to accept your manuscript in its current form for publication in Royal Society Open Science. The chemistry content of Royal Society Open Science is published in collaboration with the Royal Society of Chemistry. I apologise this has taken longer than usual.

On behalf of the Subject Editor Professor Anthony Stace and the Associate Editor Professor Tobias Hertel.

RSC Associate Editor
Comments to the Author:
(There are no comments.)

Reviewer(s)' Comments to Author:

Appendix A

Dear Editor,

I am writing to submit our revised manuscript entitled "Enhanced magnetic performance of aligned wires assembled from nanoparticles: from nanoscale to macroscale". We appreciate the valuable comments from the reviewers, which helped improve our manuscript. We have addressed these comments in the revised manuscript. In addition, point to point responses are provided below.

Reviewer 1

1.1 *According to the authors, the manuscript describes a method to produce dense arrays of nanowires (NWs) with magnetic coercivity and remanent magnetization "much higher than that of the nanoparticles (NPs)". However, this is not supported by reported data. In figure 5, hysteresis loops of array and NPs are very similar, with only a very small increase of coercivity (less than 6.5 %). The case of figure 6 is even worse: the hysteresis loops measured in to perpendicular directions of the arrays and the one measured in the NPs are essentially the same, showing no contribution from shape anisotropy in these NWs. The third figure related magnetism (figure 7), shows no change in coercivity and small increase of remanence upon applied of high magnetic field. It is quite surprising for me that the application of 0.8 T (quite a large magnetic field) produces almost no effect in the system.*

Response:

Thank the reviewer for the detailed comments on this manuscript with carefulness. We carefully made revisions according to these comments. The improper description in sections of "abstract" and "results" was corrected. "Much" was removed from the sentence of "... much higher than that of the nanoparticles". The comments related to the figures was replied in detailed as below:

- (1) The coercivity increase of the aligned α'' -Fe₁₆N₂/Al₂O₃ arrays (shown in Figure 5) indeed shows not higher than the increase for arrays synthesized from direct synthesis and complex assembly method. Owing to high cost and complex nature, these methods still cannot be adopted for large-scale production [1, 2]. This study mainly focuses on developing an environmentally friendly and feasible fabrication method to produce dense arrays for potential application of large-scale production in industry. This studied method didn't show much advance in the increase of coercivity. We added the description in the first paragraph in the revised subsection "3.2 Magnetic property of arrays" as: "Although the increase in Mr/Ms and Hc are only 34.7% and 6.4% for the dispersed α'' -Fe₁₆N₂/Al₂O₃ MNPs, respectively, the result identifies that the used fabrication method is valid in producing arrays and enhancing their magnetic properties."
- (2) The reason for that the less contribution from shape anisotropy for Fe₃O₄ array (as shown in Figure 6) was discussed in the end of the second paragraph in the same subsection as: "However, compared to the enhancement for the Fe₃O₄

array, the enhancement for the α'' -Fe₁₆N₂/Al₂O₃ arrays was much larger, owing to the higher isotropic of α'' -Fe₁₆N₂/Al₂O₃ MNPs than that of Fe₃O₄ MNPs.”

(3) In fact, Figure 7 shows a higher enhancement of magnetic properties for the spindle-shape MNPs than that in Figure 5. The coercivity increased 11.7% and 19.0% under 0.8 T and 15 T, respectively. This phenomenon was explained in the end of subsection 3.3 as: “The increase of Mr/Ms and Hc for the spindle-shaped α'' -Fe₁₆N₂/Al₂O₃ are both higher than those for the cube-like one (34.7% and 6.4%) under the same EMF of 15 T. The increased enhancement of magnetic properties is due to the spindle-shape anisotropy, ...”

1.2 *It is also remarkable that most of the hysteresis loops are measured only with the applied field parallel to the axis of the nanowires and, therefore, magnetization processes are mainly due to domain wall nucleation, pinning and movement. Under these conditions it is difficult to evaluate the shape anisotropy induced in the nanowires. It would be important to show the hysteresis loops measured with the applied field perpendicular to the wire axis. Under these conditions it is easier to measure the anisotropy field.*

Magnetic properties of nanowires are governed by shape anisotropy and many examples can be found in literature. In the case of the reported nanowires, I cannot see the expected behavior for magnetic nanowires.

Response:

We highly agree with the comment from the reviewer on the hysteresis loop measurement with the applied field perpendicular to the wire axis. We conducted the measurement for both the fabricated α'' -Fe₁₆N₂/Al₂O₃ and Fe₃O₄ arrays, as shown in Figure 5 and Figure 6, respectively. The comparative measurement reveals a large difference in magnetic property when the magnetic field applied parallel and perpendicular to arrays, as an example of Figure 5:

Figure 5. Magnetic characterization of dispersed core-shell α'' -Fe₁₆N₂/Al₂O₃ MNPs and their assembled array under an EMF of 15 T with the measured magnetic field applied parallel and perpendicular to arrays.

A table was added to the revised subsection 3.2 of summarizing the magnetic property to emphasize the shape anisotropy for the fabricated arrays as Table 1. It shows the expected behavior for the fabricated magnetic arrays.

Table 1. Summary of magnetic properties at 300 K for dispersed α'' -Fe₁₆N₂/Al₂O₃ and Fe₃O₄ MNPs and their assembled arrays under an EMF of 15 T with the measured magnetic field applied parallel and perpendicular to arrays.

Array	EMF direction	Mr/Ms (%)	Hc (kOe)
Fe ₁₆ N ₂	Particles	49	1.05
	Parallel	66	1.12
	Perpendicular	13	0.77
Fe ₃ O ₄	Particles	19	0.097
	Parallel	26	0.136
	Perpendicular	14	0.083

1.3 *In addition to these remarks, there are other issues that should be addressed. It is true that, from the XRD, it seems that the material is better oriented because there are less reflections in the diffractograms. However, the intensity of the peaks is lower and, in the particular case of the Fe16N2 array, the signal/noise ratio very poor. It seems that, when particles are oriented to form the nanowires, their crystal quality is worse. Is there any reason for these?*

Response:

The noise is likely owing to the addition of epoxy resin. The reason was discussed in the the second paragraph in the revised subsection 3.1 as: “Different from MNPs, the XRD patterns of the fabricated arrays possess a slightly uphill baseline and obvious noise on baselines. This difference likely originates from the addition of epoxy resin during the fabrication of arrays. Similar phenomena were also shown for fabricating α'' -Fe₁₆N₂ contained fibers and carbon nanotube films in our previous studies [3-5].”

1.4 *Figure 3 shows SEM images of the NWs. From the figures (in particular panels b, c and e), some texture is shown. However, when zooming (panels d and f), no evidence of NWs formation is clear, but clusters or disordered particles.*

To sum up, I believe the method proposed by the authors could be interesting, but more clear experimental proofs of the formation of nanowires from the structural and magnetic point of view should be presented.

Response:

The obtained arrays were composed by wide MNP fibers with diameter of about 1 μ m, instead of single MNP chains. The unclear panel 3d was replaced a new one in the revised manuscript as below. The appeared disordered particles were discussed in the first paragraph in the revised subsection 3.1 as: “Different from previous nanowires assembled through dipolar interactions between single MNPs and with the same diameter of single MNPs [6-8], the assembled wires in this study wasn’t single MNP chains and contained many MNPs in the cross-section of every single wire. Thus, the average diameter of the wires was larger than that of the composited MNPs. The diameters of wires, on the order of 1 μ m, are difficult to estimate because they are overlapped with each other.”

We thank the reviewer for the positive comment on our proposed method. Further experimental study is indeed required to reveal the basic formation mechanism in the future.

Reviewer 2

The use of magnetic field as a parameter, which allows a self-assembly process of magnetic nanoparticles, constitutes very interesting topic in the material science. In particular, this approach plays an important role in the synthesis of wire-like nanomaterials composed of nanoparticles. Even if the reviewed manuscript presents the quite interesting results, I do not recommend this work to be published in the Royal Society Open Science in the present version due to some factors. Few of them are listed below.

Response:

We would like to appreciate the reviewer for the positive comments and pointing out the potential impact of this study.

2.1 *The reviewed manuscript contains a lot of grammar errors and, in general, it is written using poor English. Therefore, the authors should improve all language mistakes before the new submission.*

Response:

The English language of this manuscript had been professionally edited by Elsevier Language Editing Services (Order Number: 179717). The certification was also attached for reference. The revision version of the manuscript had been corrected by Dr. Monique Teich.

2.2 *The presented experimental results are not entirely convincing. There is no explanation why the authors decided to apply the external magnetic field (EMF) of 0, 0.8 and 15 T for studying the alignment of spindle-shape core-shell α "-Fe₁₆N₂/Al₂O₃ MNPs?*

Response:

An additional description for studying the alignment of spindle-shape MNPs was added in the end of the first paragraph in the revised subsection 3.3 as:

“In a new scenario, if elongated MNPs with shape anisotropy can rotate freely in a solvent, an EMF can make them to orient along a magnetic easy axis oriented to the magnetic line and inevitably induce the formation of arrays. Thus, the shape anisotropy can increase the alignment of single-domain MNPs with magnetic easy axes under an EMF [9, 10]. The shape anisotropy will be enhanced if external magnetic field is applied to spindle-shape MNPs.”

2.3 *At the beginning, the authors present the dimensions of initial materials taken to the dispersion process. Then, they claim that this process leads to a partial damaging of α'' -Fe₁₆N₂/Al₂O₃ MNPs (comparison shown in Fig. 1) and their average dimensions become the same. How is it possible?*

Response:

Thank the reviewer for pointing out the mistake. It was corrected in the revised manuscript as:

“Average diameters after dispersion are 45.3 nm and 64.7 nm for α'' -Fe₁₆N₂/Al₂O₃ and Fe₃O₄ MNPs, respectively.”

2.4 *The explanations provided on pages 10-11, why the intensities of XRD peaks vary, are unclear. They need to be improved.*

Response:

The disadvantage of XRD peaks from the arrays can't be avoid, owing to the addition of epoxy resin. The reason was discussed in the the second paragraph in the revised subsection 3.1 as: “Different from MNPs, the XRD patterns of the fabricated arrays possess a slightly uphill baseline and obvious noise on baselines. This difference likely originates from the addition of epoxy resin during the fabrication of arrays. Similar phenomena were also shown for fabricating α'' -Fe₁₆N₂ contained fibers and carbon nanotube films in our previous studies [3-5].”

2.5 *No information is provided which the XRD database has been used in order to assign the peak positions to the Miller indices.*

Response:

Thank the reviewer for pointing out the missing information. It was added in the revised 2.3 subsection as:

“..., while assigned Miller indices of the peaks were obtained from the JCPDS database.”

2.6 *Also, the explanation provided on page 14, why the isotropic MNPs could be used for sensors, is not clear enough. This part should contain more information.*

Response:

Improper description in this part was revised to avoid misunderstanding in the revised manuscript as:

“The generated detectable magnetic field results suggest that the fabricated arrays from isotropic MNPs possess the potential application in could be used for sensors and tiny permanent magnets.”

References:

- 1 Krajewski, M. 2017 Magnetic-field-induced synthesis of magnetic wire-like micro- and nanostructures. *Nanoscale*. **9**, 16511-16545.
- 2 Scott, J. A., Totonjian, D., Martin, A. A., Tran, T. T., Fang, J. H., Toth, M., McDonagh, A. M., Aharonovich, I., Lobo, C. J. 2016 Versatile method for template-free synthesis of single crystalline metal and metal alloy nanowires. *Nanoscale*. **8**, 2804-2810.

- 3 Li, Q., Kartikowati, C. W., Ogi, T., Iwaki, T., Okuyama, K. 2017 Facile fabrication of carbon nanotube forest-like films via coaxial electrospray. *Carbon*. **115**, 116-119.
- 4 Kartikowati, C. W., Suhendi, A., Zulhijah, R., Ogi, T., Iwaki, T., Okuyama, K. 2016 Preparation and evaluation of magnetic nanocomposite fibers containing α "-Fe₁₆N₂ and α -Fe nanoparticles in polyvinylpyrrolidone via magneto-electrospinning. *Nanotechnology*. **27**, 025601-025610.
- 5 Kartikowati, C. W., Suhendi, A., Zulhijah, R., Ogi, T., Iwaki, T., Okuyama, K. 2016 Effect of magnetic field strength on the alignment of α "-Fe₁₆N₂ nanoparticle films. *Nanoscale*. **8**, 2648-2655.
- 6 Sun, Q., Wang, S. G., Wang, R. M. 2012 Well-Aligned CoPt Hollow Nanochains Synthesized in Water at Room Temperature. *J Phys Chem C*. **116**, 5352-5357.
- 7 Zhang, J. Q., Xiang, W. F., Liu, Y., Hu, M. H., Zhao, K. 2016 Synthesis of High-Aspect-Ratio Nickel Nanowires by Dropping Method. *Nanoscale Res Lett*. **11**, 118-222.
- 8 Wang, J., Chen, Q. W., Zeng, C., Hou, B. Y. 2004 Magnetic-field-induced growth of single-crystalline Fe₃O₄ nanowires. *Adv Mater*. **16**, 137-140.
- 9 Niu, H., Chen, Q., Ning, M., Jia, Y., Wang, X. 2004 Synthesis and One-Dimensional Self-Assembly of Acicular Nickel Nanocrystallites under Magnetic Fields. *The Journal of Physical Chemistry B*. **108**, 3996-3999.
- 10 Lisjak, D., Mertelj, A. 2018 Anisotropic magnetic nanoparticles: A review of their properties, syntheses and potential applications. *Progress in Materials Science*. **95**, 286-328.

Appendix B

Dear Editor,

I am writing to submit our revised manuscript entitled "Enhanced magnetic performance of aligned wires assembled from nanoparticles: from nanoscale to macroscale". We appreciate the valuable comments from the reviewers for the second review, which helped improve our manuscript. We have addressed these comments in the revised manuscript. In addition, point to point responses are provided below.

Reviewer 1

After the revision, the manuscript has been improved, in particular the part related to the formation of nanowires and the study of the structure. However, the analysis of the magnetic properties is very weak and the authors have not fully addressed the previous concerns.

So, to conclude, as mentioned before, I believe the manuscript has potential interest. However, I believe authors should center the manuscript in the fabrication and structural characterization, and rewrite the magnetic properties to support the formation of real wires and not agglomeration of particles.

Response:

We would like to sincerely appreciate the reviewer for the positive comments and further comments to improve our manuscript. The further comments are carefully addressed as below.

1.1 *In particular, I suggested measuring with magnetic field applied in the direction perpendicular to the wires to determine the anisotropy field. This parameter is important to determine whether the magnetic properties come from the wire structure or not. This parameter has not been calculated.*

I agree that figure 5 reflects, probably, the shape anisotropy in the magnetic properties due to the wire formation, although it is not very clear in the text. However, the authors do not explain why, in figure 6, all hysteresis loops are basically identical, with very small variation. This, in fact, is an indication of the existence of decoupled particles and not a wire. This, in fact, is also shown in figure 3f: there is no wires but chains of decoupled NPs.

Response:

Thanks to this valuable comment on the anisotropy field, we made the calculation in the revised manuscript regarding to the comment. The value of anisotropy field was calculated from the measurement with magnetic field applied in the direction perpendicular to the wires was conducted for the fabricated α'' -Fe₁₆N₂/Al₂O₃ and Fe₃O₄ arrays, as shown in Figures 5 and 6. The values and description was added in the first and second paragraph in the subsection 3.3 as: 'The anisotropy field, calculated from the measured easy axis (parallel to arrays) and hard axis (perpendicular to arrays) loops of the reduced magnetization [1], is 6.235 kOe for the α'' -Fe₁₆N₂/Al₂O₃ arrays.' 'The anisotropy field for the Fe₃O₄ arrays is 0.202 kOe, which is much lower than that of the α'' -Fe₁₆N₂/Al₂O₃ arrays.'

Since the Fe₃O₄ array shows less magnetic anisotropy than the α'' -Fe₁₆N₂/Al₂O₃ array, as indicated by the difference of the anisotropy field (6.235 kOe >> 0.202 kOe). Thus there is less difference for the hysteresis loops with magnetic field applied in the

directions perpendicular and parallel to the Fe_3O_4 arrays, as well as less coupled NPs in Figure 3(f). This description was added in the second paragraph in the subsection 3.2 as: ‘Thus there is less difference for the hysteresis loops with magnetic field applied in the directions perpendicular and parallel to the Fe_3O_4 arrays than that to the $\alpha''\text{-Fe}_{16}\text{N}_2/\text{Al}_2\text{O}_3$ arrays, as well as less coupled NPs in Figure 3(f).’

1.2 *Authors talk about magnetic isotropy in several parts of the manuscript. Do they refer to magnetic anisotropy?*

Response:

The mistake was corrected in the revised version with the replacement of ‘isotropy’ with ‘anisotropy’.

1.3 *Finally, with the reported values of M_r/M_s and H_c , I cannot see how this material can be used as permanent magnets.*

Response:

The inaccurate description was revised with removing ‘permanent’.

Reviewer 2

I was asked again to review the re-submitted version of manuscript entitled ‘Enhanced magnetic performance of aligned wires assembled from nanoparticles: from nanoscale to macroscale’. My opinion about the topic of the performed investigations does not change. I still think that the use of magnetic field as a manufacturing parameter is a very interesting approach to obtain the anisotropic magnetic wire-like materials.

The new version of manuscript is more clear than the previous one and the results are presented in a more convincing manner. The previously mentioned lapses are well addressed. Therefore, I would like to recommend this work to be published in the Royal Society Open Science after minor revision. The list of remarks is included below.

Response:

We would like to appreciate the reviewer for the positive comments and pointing out the potential impact of this study. The list of responses is addressed as below.

2.1 *Again, I need to point out the lack of explanation why the Authors have decided to apply such a high external magnetic field (15 T) in order to obtain the wires of $\alpha''\text{-Fe}_{16}\text{N}_2/\text{Al}_2\text{O}_3$ MNPs. In general, they have presented that the much weaker magnetic field of 0.8 T is not sufficient for assembling highly-anisotropic material but between 0.8 T and 15 T is a high difference. Could the Authors comment this issue?*

Response:

There was only external magnetic field with 0.8 T and 15 T available for the experiment in the lab. This information was added in the revised manuscript as: ‘..., which was the only dependable and available one in the laboratory.’

2.2 *The second concern is associated with the sentence included in abstract. The Authors claim that “(...) This study reports a facile method to fabricate arrays of magnetic wires (...)”. I agree that the concept of the manufacturing process is rather facile but the performance of this process might not be so easy due to the application of very high magnetic field of 15 T. Could the Authors comment this issue?*

Response:

We agree with the comment on the requirement of strong magnetic field to fabricate magnetic arrays at high quality. The last sentence on its implication was revised accordingly as: ‘**Although high intensity external field benefits for the fabrication of magnetic arrays, the newly developed technique provides an environmentally friendly and feasible approach to fabricate magnetic wires in highly dense arrays in open environment conditions.**’

2.3 *Could the Authors comment why the values of saturation magnetizations measured for the magnetic field applied parallel and perpendicular to the array of wires are different?*

Response:

Thank the reviewer for pointing out the improper expression, and it was removed from previous pages 14 and 17. The Ms in the parallel direction of the array was slightly higher than that in the perpendicular direction, owing to the slightly damage during the operation during SQUID measurement.

2.4 *Also, I found three mistyping in the re-submitted manuscript.
page 9 line 44; instead of ‘weren’t’ should be ‘were not’
page 19 line 24; instead of ‘Table 1’ should be ‘Table 2’
page 20 line 10; missing unit right behind ‘1000’*

Response:

Thank the reviewer very much for pointing out the mistyping. The first two ones were corrected accordingly, while there is no unit for the aspect ratio of ‘1000’.

Reference:

1 Neudert, A., McCord, J., Schafer, R., Schultz, L. 2004 Dynamic anisotropy in amorphous CoZrTa films. *J Appl Phys.* **95**, 6595-6597. (10.1063/1.1667796)